# Stocking Density Affects Welfare Indicators in Horses Reared for Meat Production

**DOI:** 10.3390/ani10061103

**Published:** 2020-06-26

**Authors:** Federica Raspa, Martina Tarantola, Domenico Bergero, Claudio Bellino, Chiara Maria Mastrazzo, Alice Visconti, Ermenegildo Valvassori, Ingrid Vervuert, Emanuela Valle

**Affiliations:** 1Department of Veterinary Sciences, University of Turin, 10095 Grugliasco, Italy; martina.tarantola@unito.it (M.T.); domenico.bergero@unito.it (D.B.); claudio.bellino@unito.it (C.B.); chiara.mastrazzo@edu.unito.it (C.M.M.); alice.visconti@edu.unito.it (A.V.); emanuela.valle@unito.it (E.V.); 2Public Veterinary Service, ASL TO5 Piedmont, Italy; e.valvassori@tin.it; 3Institute of Animal Nutrition, Nutrition Diseases and Dietetics, Faculty of Veterinary Medicine, Leipzig University, 04103 Leipzig, Germany; ingrid.vervuert@vetmed.uni-leipzig.de

**Keywords:** welfare, horse, meat production, stocking density, feeding management

## Abstract

**Simple Summary:**

Not enough effort is being made to safeguard the welfare of horses reared for meat production. These horses are kept in intensive breeding farms where they are housed in group pens at high stock densities and fed high amounts of concentrates. The aim of the study is to evaluate whether the stocking density of horses raised in group pens for meat production and their feeding management affects their welfare according to different stocking density. According to our results, when the horses had more than 4.75 m^2^/horse, many indicators were affected (i.e., improvement of coat cleanliness, improvement of bedding quantity, improvement of mane and tail conditions, less resting in a standing position, and less feeding related to the greater space available at the feed bunk). However, a further increment of space and/or changes in management regimes may be necessary to improve all the welfare indicators. The results also revealed the need to improve the feeding management of these animals.

**Abstract:**

Horses kept for meat production are reared in intensive breeding farms. We employed a checklist adapted from the Animal Welfare Indicators (AWIN) assessment protocol. Our evaluation aims to assess whether welfare indicators are influenced by stocking densities (m^2^/horse) and feeding strategies applied. An analysis was carried out on the data obtained from 7 surveys conducted at a single horse farm designed for meat production. In each survey, the same 12 pens were assessed, but on each occasion, the horses in the pens had been changed as had the stocking densities. Briefly, 561 horses aged 16 ± 8 months (mean ± standard deviation) were evaluated. Two stocking density cut-off values (median and 75th percentile: 3.95 and 4.75 m^2^/horse, respectively) were applied to investigate the effect of stocking density on horse welfare. Data were analysed using Mann–Whitney U and Fisher’s exact tests (*p* < 0.05). When cut-off was set as the median percentile, lower stocking density was associated with improvements in body condition score (BCS), coat cleanliness and bedding quantity, less coughing, less resting in a standing position, and less feeding related to the greater space available at the feed bunk. When the 75th percentile cut-off was used, indicators that improved were coat cleanliness, bedding quantity and mane and tail condition, as well as less resting in standing position and less feeding related to the greater space available at the feed bunk. Accordingly, the use of two different stocking density cut-off values showed that the increase of space allowance affected specific welfare indicators. Further increment of space and/or changes in management regimes should be investigated to improve all the indicators. Moreover, results related to feeding indicated the need to intervene as starch intakes exceeded recommended safe levels, negatively affecting horse welfare.

## 1. Introduction

Faostat data [1] indicate that more than half a million horses are slaughtered in Europe each year for meat production. In the past, most horse meat was derived from the slaughter of horses at the end of their working lives, whereas, nowadays, horse meat is mainly obtained from the specific breeding of heavy draft breeds [2]. According to Tateo et al. [3], farms breeding horses for meat primarily rear young horses. To increase their meat production performances, these farms apply intensive farming systems. However, concerns about animal welfare related to overcrowding and intensive feeding regimes have been raised over intensive farming systems [4]. High-density group housing can negatively affect horse welfare, influencing both their health and behaviour [5]. Moreover, in order to reduce the length of the fattening period and obtain fast increases in body weight, breeders often feed the animals with a high-starch diet. However, it is well known that feeding horses with high amounts of concentrates can negatively affect their intestinal health, increasing the risk for colic and gastrointestinal disorders [6].

Several studies have underlined the negative effects of high stocking density on the welfare of livestock species [7,8,9]. However, to the best of our knowledge, no research today has focused on horses farmed for meat production. Few studies have evaluated the effects of space allowance on the welfare of horses, and they are mainly based on some behavioural or physiological aspects [10,11,12,13]. However, stocking density is recognised as crucial to reaching an adequate level of welfare at farm level. [14]. The general approach of the European Union (EU) to ensure farm animal welfare is to increase the space allowance per animal [15]. Accordingly, the minimum space requirements in group housing systems have been set for pigs [16], poultry [17], and cattle [18]. However, no specific EU Directives are defined for meat production horses [19]. The first indications about minimum space requirements for horses housed in group pens have been provided by the Swiss Federal Council in the Animal Welfare Ordinance (TSchV) of 23 April 2008 [20]. In this document, the minimum space allowance per horse is based on the withers height of the individual group members. According to Burla et al. (2017) [11], these minimal requirements are not based on scientific evidence and may not be adequate to guarantee adequate welfare for all horses of a given group [11]. At the European Union level, this criterion was then adopted in the Animal Welfare Indicators (AWIN) welfare assessment protocol for horses [21].

The AWIN protocol is based on the assessment of animal-based indicators and follows the Welfare Quality^®^ approach that consists of four welfare principles and twelve welfare criteria [22]. The four welfare principles are good feeding, good housing, good health, and appropriate behaviour. They represent the founding elements of the Five Freedoms [23] since they describe the needs of animals that should be satisfied in order to cover all aspects of animal welfare [22]. However, according to Mellor [24], affective outcomes are not sufficiently addressed in the Welfare Quality^®^ system, being addressed only briefly in the list of welfare criteria. Indeed, the four welfare principles are primarily structured to evaluate specific physical/biological functions, so they have a predominantly physiological orientation. However, as Mellor discusses [24], it is necessary to identify how physical/biological imbalances can influence the affective state and, consequently, the welfare of the animals. For this reason, Mellor proposes the Five Domains model that includes the fifth “mental” domain, the aim of which is to evaluate the animals’ mental state.

The study of animal welfare requires a multidimensional approach that involves the examination of a panel of welfare indicators encompassing all components of animal welfare [25]. Accordingly, welfare assessment involves three categories of indicators: resource-based, management-based, and animal-based [26,27]. Some criticisms have been made regarding the application of protocols built on animal-based indicators due to the difficulty in applying them at the farm level—the protocols being very time-consuming and costly [28]. Indeed, animal welfare is not an easy subject to study, and identifying the best protocol to apply on any given farm is difficult. The European Commission has financed the development of a specific protocol that considers animal-based indicators to assess and promote horse welfare—the AWIN protocol—presently the only tool validated by the European Commission for the assessment of equine welfare.

However, when the aim is to assess equine welfare on farms geared towards meat production, some limitations of the AWIN protocol become evident. As clearly underlined in the section dedicated to horses housed in groups pens, the AWIN protocol still needs to be refined and improved in light of the results of up-to-date scientific research on horses reared in this manner. Moreover, the AWIN protocol was developed in relation to horses aged 5 years or older. As such, it is imperative that this tool is revised for its use on intensive breeding farms that rear young horses (less than 5 years old) in high-density group pens.

In the present study, a checklist adapted from the AWIN protocol and based on the Welfare Quality^®^ principles was developed to evaluate whether the welfare indicators selected were influenced by the main causes of concern that regard intensive breeding farms: stocking density and feeding management. We hypothesise that welfare would be poorer at higher stocking densities and that some welfare indicators could be negatively affected by the feeding strategies adopted with meat production in mind. We tested the effect of two different stocking density cut-off values (the median and 75th percentile values), dividing the data into two groups (low vs. high stocking densities) to assess whether any improvements in horse welfare could be observed with even a small increase in space allowance per horse.

## 2. Materials and Methods

The present study was approved by the Ethical Committee of the Department of Veterinary Sciences of the University of Turin (Italy, Prot.n.2202, 8/04/2019). It was conducted in the presence of representatives of the Regional Veterinary Services. The owner of the horses agreed to the purpose of the study.

### 2.1. Data Collection

The welfare assessment was carried out on the biggest horse breeding farm for meat production in northern Italy. Seven surveys were conducted between April and June. The surveys commenced two hours after the morning meal and lasted approximately 3.5 h (from 9:00 a.m. until 12:30 p.m.).

The farm in question adopts intensive farming methods and, at any one time, houses around 300 young horses belonging to different breeds of both sexes—colts (not gelded) and fillies aged 16 ± 8 months (mean ± standard deviation (SD)). It sends a total of 2000 animals to slaughter each year. The horses were housed in group pens situated in a barn with two open sides; they had no access to any outdoor paddock area. Pens were enclosed by horizontal metal rail bars, which also delimited the pens at the feed bunk level. One automatic drinker providing tap water was available in each pen independent of the number of the animals enclosed. The floor was concrete and covered with barley straw bedding once a day before the evening meal by an automatic straw-dispersing tractor programmed to cover the entire pen floor with a thickness of at least 15 cm of straw. The number of animals per pen varied, and male and female horses were not separated. Horses were not fed on an individual basis; instead, twice a day (7:00 a.m. and 6:00 p.m.), each pen was provided with long stem self-produced meadow hay (approximately 6 kg/animal/day) and an amount of pelleted feed equal to 8 kg/animal/day. The pelleted feed was a cereal-based commercial feed (complementary feed; labelled to contain crude protein 14.50%, ether extract 3.50%, crude fibre 5.70%, ash 6.60%; as fed: starch 55%).

The farm contained a total of 24 pens; of these, every second pen was selected for assessment, providing a total of 12 pens for evaluation by means of the seven surveys conducted over the study timespan. At the time of each survey, the horses in each group pen had changed, as had the number of animals it contained. As such, different stocking densities could be evaluated by means of the welfare assessment checklist. Table 1 reports the physical characteristics of the 12 selected pens, and the median and 25th–75th percentile values regarding the number and height of the horses housed within each pen for the seven surveys conducted.

### 2.2. Welfare Assessment Checklist

A checklist adapted from the AWIN welfare assessment protocol for horses [21] was employed by five equine veterinarians who are experts on welfare protocols. Before starting the study, the evaluators received specific training on the welfare checklist, and at the end of the training period, interobserver reliability was evaluated, as indicated in the statistical analysis section.

Table 2 shows the welfare assessment checklist developed and used by the evaluators. Each evaluator independently filled out his/her own checklist. The checklist contained four sections, each regarding one of the four welfare principles of the Welfare Quality^®^ approach: good feeding, good housing, good health, and appropriate behaviour. Horse welfare was assessed according to the welfare criteria and welfare indicators belonging to each principle. The welfare indicators included resource-based, management-based, or animal-based indicators and are written in bold font in the following sections.

#### 2.2.1. Good Feeding

The welfare principle “good feeding” was described by its two welfare criteria: “appropriate nutrition” and “absence of prolonged thirst”.

To assess “appropriate nutrition”, the body condition score (BCS) was rated and recorded. The BCS is the only welfare indicator used in the AWIN protocol to describe the welfare criteria “appropriate nutrition”. It is scored using a 5-point scale [31] in which the nutritional status of an animal is assessed through observation and palpation of anatomical key areas. In the present study, the BCS of the horses was scored as “thin”, “normal” or “fat” by means of the visual appraisal of the animals’ shape alone, since it was not possible to touch the horses during the assessment (see Table 2, with associated guidance notes and illustrative photographs). The number of horses per pen judged as “thin” was recorded and used in the statistical analysis. This study also considered space allowance at the feed bunk as a welfare indicator of “appropriate nutrition” since easy access to feed troughs must be guaranteed to ensure the welfare of animals in production systems [32]. Space allowance at the feed bunk (m/horse) was calculated by dividing the length of the feed bunk (meters) by the number of horses within the pen.

The welfare criterion “absence of prolonged thirst” was assessed by considering water availability and water point cleanliness. Water availability was assessed by evaluating the correct functioning of the automatic drinkers. Water point cleanliness was scored as suggested by the AWIN protocol; specifically, the drinkers were scored “dirty” if both the bowl and water were dirty (i.e., the presence of organic materials, such as feed, soil or faeces); “partly dirty” if the bowl was dirty but the water clean, or “clean” if both bowl and water were clean (see Table 2 with associated guidance notes and illustrative photographs). The frequency (%) of the automatic drinkers scored as adequate or inadequate was calculated and used in the statistical analysis.

#### 2.2.2. Good Housing

The welfare principle “good housing” includes the welfare criteria “comfort around resting”, “thermal comfort” and “ease of movement”.

Comfort around resting was evaluated by considering the two welfare resource-based indicators, “bedding quantity” and “bedding cleanliness”, as used in the AWIN protocol, plus “coat cleanliness”.

The AWIN protocol scores the former two indicators in a qualitative manner only through the use of pictures. Here, in order to achieve a more standardised method, we developed a specific scoring system to evaluate bedding quantity and cleanliness. Bedding quantity was scored as adequate when ≥70% of the floor was covered (defined in the AWIN protocol as “sufficient bedding material”), and inadequate if >30% of the floor was not covered (defined in the AWIN protocol as “no bedding material” and “insufficient bedding material”; see Table 2 with its detailed guidance notes and photographs illustrating the scores). Bedding cleanliness was scored as adequate if ≥70% of the bedding was clean (defined in the AWIN protocol as “clean bedding material”) and inadequate when >30% of the bedding was dirty (defined in the AWIN protocol as “dirty bedding material”; see Table 2 with its detailed guidance notes and photographs illustrating the scores). For the statistical analysis, bedding quantity and bedding cleanliness were expressed as frequencies (%) of scores.

Coat cleanliness was also taken into consideration for the assessment of “comfort around resting”. We decided to evaluate this welfare indicator as it reflects the environmental conditions in which the animals are kept. A specific 5-point scoring system was designed to assess coat cleanliness (see Table 2 with its detailed guidance notes and photographs illustrating the scores). Horses were assigned a score of 1 if they were completely dirty; a score of 2 if they presented dirty limbs, abdomen, barrel, flanks and neck; a score of 3 for dirty limbs, and abdomen; a score of 4 for dirty limbs only; a score of 5 for a completely clean horse. A coat cleanliness score of 1, 2 or 3 was rated “dirty”. The number of horses per pen rated as dirty was used for the subsequent statistical analysis.

For the welfare criterion “thermal comfort”, since it was not possible to evaluate this parameter by examining whether the animals that showed clinical signs of thermal stress, as suggested in the AWIN protocol, thermal comfort was instead evaluated through the measurement of environment temperature (°C) and relative humidity (%). These measurements were taken in front of each pen using a digital thermometer and hygrometer. According to the Wageningen UR Livestock Research Welfare Monitoring System [30], the temperature was considered adequate when it was within the horse’s thermoneutral zone (+5 °C to +25 °C); and relative humidity was deemed to be adequate when the values ranged from 60% to 80%.

The welfare criterion “ease of movement” should regard the quality of the exercise horses are able to partake in. The AWIN protocol describes this management-based indicator by referring to the possibility for animals to spend part of their day performing activities in outdoor areas. Since it was not possible to apply this welfare indicator in the evaluation of animals kept in a production system, we decided to evaluate each pen’s area (m^2^) and stocking density (m^2^/horse) to gain some data pertaining to the animals’ possibility for “ease of movement”. Once the area of a pen was calculated, it was then divided by the median height of the horses, measured to the withers, within the pen. As we were not able to touch the animals, a laser meter was used to measure the height of animals at the withers. Measurements were conducted for the tallest and the shortest horse in order to ascertain the height range for the horses within a pen. The measurement was made at the moment in which the animal was standing in a position that was parallel to the wall or to the horizontal metal rail bars. The stocking density was considered adequate or inadequate according to the indications provided in the AWIN protocol in the section adapted for group-housed horses [21]. Accordingly, if animals were assessed to measure between 120 cm and 148 cm at the withers, a minimal space of 7 m^2^/horse was required, whereas if the heights ranged between 148 cm and 162 cm, an adequate space allowance should not be less than 8 m^2^/horse.

#### 2.2.3. Good Health

The welfare principle “good health” includes three welfare criteria: “absence of injuries”, “absence of diseases”, and “absence of pain and pain induced by management procedures”.

“Absence of injuries” is described by evaluating the animal-based indicators “presence of integument alterations” and “presence of swollen joints—signs of lameness”, as well as “mane condition” and “tail condition”.

The presence of integument alterations was evaluated by recording the extent of visible areas of alopecia, skin lesions (as superficial or deep wounds), tumefaction, and swelling. Since it was not possible to approach the animals, a visual inspection of the body of each animal was performed. In the checklist, the number of horses within each pen presenting at least one visible integument alteration was recorded and used for statistical analysis.

The number of horses presenting visibly swollen joints and/or signs of lameness was recorded. In addition, a visual inspection of the body of each horse within the pen was performed, focusing attention on the distal limbs, the shape of the hooves, and the animals’ movements.

In our assessment of the welfare criterion “absence of injuries”, we decided to introduce two additional animal-based indicators on the basis of their initial observations of the animals; they were mane condition and tail condition. We decided to include these welfare indicators as they seemed to reflect the specific housing and management features of this kind of farm. In particular, the observation of alterations to the mane and/or tail seemed to constitute a specific “occupational ailment” in this specific context. A specific 3-point scoring system was defined for both mane and tail condition: a score of 1 indicated good mane/tail condition for their entire length; a score of 2 indicated areas of broken and/or absent mane/tail hair, but with the skin intact; and a score of 3 indicated a damaged mane/tail with areas of broken and/or absent mane or tail hair and injured skin (see Table 2 with its detailed guidance notes and the photographs illustrating the scores).

The welfare criterion “absence of diseases” was assessed using four animal-based welfare indicators: “coughing”, “abnormal breathing”, “discharges”, and “consistency of faeces”. Coughing and abnormal breathing were recorded as the number of horses presenting these symptoms. To evaluate breathing, the head and the flanks of each horse were observed. Breathing was considered abnormal when at least one of the following clinical signs were observed: flaring of the nostrils, extended head and neck, increased respiratory rate, or asynchrony between movements of the chest and the abdomen. The number of horses within the group pen coughing or with abnormal breathing was recorded and used in the statistical analysis. Nasal and ocular discharges were evaluated by observation. This assessment was performed at the same time as the assessment for coughing and abnormal breathing. Once again, the number of horses within the group pen presenting these clinical signs was recorded.

The consistency of faeces was considered by evaluating the shape of the faeces present in the bedding of each group pen and recorded as normal and/or abnormal. Faeces were scored as abnormal if the shape of the faeces was not conserved. For statistical analysis, the frequency (%) of group pens containing abnormal faeces was calculated.

To assess the welfare criterion “absence of pain and pain induced by management procedures”, the indicator “state of awareness” was evaluated. The AWIN protocol recommends the use of the Horse Grimace Scale that assesses equine facial expressions for the assessment of pain; however, this was not deemed feasible in the present study, thus the concept of state of awareness was introduced as an alternative. This involved observing the animals and noting whether they presented any symptoms of an “abnormal” state of awareness, which includes the adoption of a depressed or an alarmed stance, paying no attention to the surrounding environment and an inadequate response to stimuli, such as light, noise and the presence of people. The number of horses per pen that presented an abnormal state of awareness was recorded and used in the statistical analysis.

#### 2.2.4. Appropriate Behaviour

To assess the welfare principle “appropriate behaviour”, the following welfare indicators were considered (as measures of the welfare criteria “expressions of social behaviour” and “expressions of other behaviours”): feeding, watching, mutual grooming, resting in a standing position, resting in a lying position, playing, sexual behaviours, aggressive behaviours, and stereotypic behaviours (licking, crib-biting, weaving, head nodding, wood chewing; see Table 2 with its detailed guidance notes). To assess these indicators, all five evaluators simultaneously observed the horses within a single pen. They were positioned at different positions outside the pen at a maximum distance of 5 m from the horses. The welfare assessment started 5 min after the placement of the evaluators, who remained still and quiet to allow the horses to become accustomed to their presence. A methodology was adapted that involved observing the horse situated the furthest to the left in the pen, then moving to the animal situated to its right, and so on. The number of horses displaying each specific behaviour was recorded and used for statistical analysis.

### 2.3. Statistical Analysis

For analytical purposes, the data pertaining to the individual group pens were assigned to one of two groups on the basis of their stocking density (m^2^/horse). The median stocking density was calculated in order to divide the data into two groups, depending on whether they were housed at a low stocking density (LSD^50th^; i.e., at or above the 50th percentile) or a high stocking density (HSD^50th^; below the 50th percentile). The 75th percentile value was also calculated, and the animals again divided into low or high stocking density groups depending on whether they were housed at or above, or below the 75th percentile stocking density (LSD^75th^ and HSD^75th^, respectively).

Data were analysed using IBM SPSS^®^ Statistics 21.0 software (SPSS Inc., Chicago, IL, USA) to identify any differences between the groups divided according to the stocking density cut-off values. The Shapiro–Wilk test was used to assess whether the data were distributed according to a normal distribution. Since the data were not normally distributed, the Mann–Whitney U and the Fisher’s exact tests were applied. A *p*-value < 0.05 was considered significant to infer that differences between the groups were related to the stocking density.

The interobserver reliability of the expert evaluators in their assessment of welfare indicators was evaluated by means of the Cohen’s kappa coefficient (Κ).

Dichotomous variables (bedding cleanliness, bedding quantity, consistency of faeces, water point cleanliness) were expressed as frequencies (% of group pens). The other welfare indicators (i.e., the nondichotomous variables) were expressed as the number (N) of horses within each group pen presenting a specific score or health condition or performing a specific behaviour.

## 3. Results

A total of 561 horses were evaluated. The horses belonged to Italian or French heavy draft breeds, and the mean age (± SD) was 16 (± 8) months.

The median values (plus 25th–75th percentiles) for environment temperature (°C) and relative humidity (%) over the seven surveys were 13 °C (11–23 °C) and 73% (55–75%), respectively.

The Cohen’s kappa coefficients (Ks) for interobserver reliability ranged between 0.61 and 1, indicating substantial (K = 0.61–0.80) to strong (K = 0.80–1) agreement between the expert evaluators.

### 3.1. Results Considering the Median Cut-Off Value for the Stocking Density

Table 3 shows the results of the Mann–Whitney U-test and Fisher’s exact tests. The median cut-off value for the stocking density was calculated in order to divide and compare the survey data according to whether the horses were housed at a low stocking density (LSD^50th^) or a high stocking density (HSD^50th^). The median cut-off value for the stocking density (m^2^/horse) was 3.95 m^2^/horse (LSD^50th^ group ≥3.95 m^2^/horse vs. HSD^50th^ group <3.95 m^2^/horse).

When the two groups were compared on the basis of the median stocking density cut-off value, significant differences were found in two of the welfare indicators of good feeding: the space at the feed bunk (m/horse; *p* < 0.001) and the BCS (*p* = 0.004). The ideal feeding space per horses at a feed trough is reported to be 1 m/horse [29]; the median space (plus 25th–75th percentiles) revealed here was 0.95 (0.70–1.30) m/horse for the LSD^50th^ group and 0.6 (0.42–0.79) m/horse for the HSD^50th^ group. Moreover, the median number of horses within the group pens scored as thin was higher for the horses in the HSD^50th^ group at 0.5 (0–2.25) compared with 0 (0–0) for the LSD^50th^ group.

Considering the welfare principle of good housing, the welfare indicators “coat cleanliness” and “bedding quantity” were shown to be influenced by the stocking density. The median number of animals scored as having a dirty coat (coat cleanliness score of 1 to 3) was lower (3, 1–4) in the LSD^50th^ group than in the HSD^50th^ group (5, 2–7) (*p* = 0.004). Therefore, a higher stocking density was associated with a significantly higher number of horses scored as having a dirty coat. The frequency (%) of pens scored as having an inadequate quantity of bedding was 56.8% in the LSD^50th^ group and 83.3% in the HSD^50th^ group (*p* = 0.021), revealing that when horses were housed at higher densities, a significantly higher percentage of pens had inadequate amounts of bedding material covering the pen floor.

For the welfare principle of good health, just one welfare indicator was affected by stocking density: the median number of horses with a cough was significantly lower in the LSD^50th^ group than in the HSD^50th^ group (*p* = 0.028).

Finally, with regard to the welfare principle of appropriate behaviour, two indicators were affected by stocking density: feeding behaviour and resting in a standing position. The median number of horses exhibiting feeding behaviour at the moment of the observation was significantly higher in the HSD^50th^ group (5, 2–6.75) than the LSD^50th^ group (2, 0.5–4) (*p* = 0.001). This suggests that, on the farm in question, horses housed at a higher stocking density are more likely to express feeding behaviour. Moreover, with regard to resting in standing position, more animals were found to express this behavior in the HSD^50th^ group (1, 0–3) than LSD^50th^ (0, 0–2) (*p* = 0.012).

### 3.2. Results Considering the 75th Percentile Cut-Off Value for the Stocking Density

The data were reanalysed by considering a stocking density (m^2^/horse) cut-off value equal to the 75th percentile: 4.75 m^2^/horse. This analysis was performed to assess whether a small increase in space allowance per horse would lead to any improvements in horse welfare. Therefore, animals in the LSD^75th^ group had a space allowance ≥4.75 m^2^/horse, whereas those in the HSD^75th^ group had <4.75 m^2^/horse. The results of the Mann–Whitney U test and the Fisher exact tests are shown in Table 4.

Considering the welfare principle of good feeding, once again, a significant difference was shown in relation to space at the feed bunk (m/horse; *p* < 0.001). When we consider a lower stocking density, this automatically correlates with a larger feeding space per animal at the feed bunk. In fact, the median space at the feed bunk was 1.3 (1.10–1.54) m/horse in the LSD^75th^ group vs. 0.70 (0.45–0.84) m/horse for the HSD^75th^ group.

Moving on to the welfare principle of good housing, the data regarding coat cleanliness and the bedding quantity were again found to differ significantly between the low and high stocking density groups. The median number of animals scored to have a dirty coat (cleanliness score of 1 to 3) was higher (4, 2–7) in the HSD^75th^ group than in the LSD^75th^ group (2, 1–4) (*p* = 0.005). The frequency (%) of pens scored as having an inadequate quantity of bedding was significantly lower (44.48%) in the LSD^75th^ group compared with the HSD^75th^ group (78.2%) (*p* = 0.016).

For the welfare principle of good health, both mane condition and tail condition were significantly influenced by stocking density when defining the groups by the 75th percentile cut-off, with *p*-values of 0.038 and 0.024, respectively. The median number of horses presenting a ruined mane (score of 3) was significantly higher (5, 3–8) in the HSD^75th^ group than the LSD^75th^ group (3.5, 3–4.75) (*p* = 0.038). Moreover, the median number of horses presenting a ruined tail (score of 3) was lower (0, 0–1) in the LSD^75th^ group than in the HSD^75th^ group (1, 0–3) (*p* = 0.024).

Considering the welfare principle of appropriate behaviour, the median number of horses expressing feeding behaviour at the moment of the welfare assessment was higher in the HSD^75th^ group (3, 2–6) than in the LSD^75th^ group (1.5, 0–3.25) (*p* = 0.002). A significant difference between groups was also found for the median number of horses standing in a resting position, which was higher in the HSD^75th^ group (1, 0–3) than the LSD^75th^ group (0, 0–0.25) (*p* = 0.003).

Interestingly, in contrast with the previous statistical analysis in which the median stocking density was used as the cut-off value, no statistical significance was shown for BCS or the presence of a cough when groups were compared on the basis of the 75th percentile cut-off value.

## 4. Discussion

The present study provides some information about the welfare status of horses reared for meat production and identifies some problems regarding this kind of intensive breeding system. The aim of the study was to test the hypothesis that stocking density and feeding management affect welfare indicators in horses reared for meat production. We decided to apply two different cut-off values when dividing the data according to stocking density to evaluate whether any improvements in horse welfare could be observed with even a small increase in space allowance per horse.

The assessment of animal welfare is a multidimensional and complex procedure that should include a combination of resource-, management- and animal-based indicators to describe the various aspects of animal welfare [33,34]. In the present study, we applied a welfare assessment checklist based on the AWIN structure. The AWIN protocol was specifically proposed and financed by the European Commission as an equine welfare assessment tool. However, as reported by AWIN, the protocol was developed for adult horses and may be difficult to apply to horses housed in group pens. As suggested by AWIN, the AWIN protocol needs to be redefined in light of up-to-date scientific research on horses kept in group pens and the specific breeding system being applied. To date, no studies have been published on the welfare of horses specifically bred on farms for meat production. As reported in the results section, the horses in the farming system studied were very young (16 ± 8 months, ±SD) and housed in groups, making it difficult to evaluate certain welfare indicators. In the present study, the key structure of the AWIN protocol was used, but certain adaptations were made to take into consideration the specific conditions of horse farming applied in this context (high stocking densities and an intensive feeding management regime).

Stocking density and adequate space allowance for horses housed in groups constitute one of the main welfare concerns regarding intensive horse farming for meat production [5]. The opportunity for movement is known to play an important role in equine welfare, having a positive effect on both physical and mental health [35]. Therefore, increasing the space allowance per horse is likely to form an important measure able to improve welfare. In the present study, the stocking density of group pens was calculated as m^2^ available per horse (m^2^/horse). Two stocking density cut-off values were calculated and considered in the statistical analyses, calculated as the median (3.95 m^2^/horse) and the 75th percentile (4.75 m^2^/horse) values. Recommendations relating to the minimum space needed per horse housed in groups are provided by the AWIN protocol that takes into consideration a horse’s height, as measured at the withers. In the present study, the height of the horses within the group pens ranged between 120 and 160 cm. According to AWIN, horses in this height range require at least 7 m^2^/horse. None of the pens at the farm provided this amount of space per horse. The use of two different stocking density cut-off values enabled us to show that an increase of just 0.80 m^2^/horse (3.95 to 4.75 m^2^/horse) was able to have a significant effect on specific welfare indicators.

Considering the welfare principle of good feeding, we revealed a significant influence of the space at the feed bunk (m/horse) at both cut-off values. The median feeding space per horse at the feed bunk was always less than 1 m/horse—the minimal distance recommended in the Code of Practice for the Care and Handling of Equines [29]—when the stocking density cut-off value was set to 3.95 m^2^/horse. Adequate feeding space per horse at the feed bunk is important in order to mimic physiological feed intake behaviour and limit competition for resources [5,32]. Under natural conditions, horses live in herds and generally forage at the same time [36], preferring to maintain a distance of at least 2 m from each other [37]. In the HSD^50th^ condition of the present study, where the median space at the feed bunk was just 0.6 m/horse, the number of horses scored as “thin” according to the BCS scoring system was significantly higher than in the LSD^50th^ condition, where the feed bunk space was closer to 1 m/horse (0.95 m/horse). This difference was no longer present when the median feed bunk space of both groups exceeded 1 m/horse (i.e., when the 75th percentile cut-off was applied). It is interesting to notice that the number of animals exhibiting feeding behaviour was always greater in the high stocking density groups, independent of which cut-off value was used (HSD^50th^ or HSD^75th^). This shows that both feed bunk space and stocking density can reciprocally influence feeding behavior.

For the welfare principle of good housing, we found stocking density to have a significant influence upon coat cleanliness. The number of animals rated as having a dirty coat was consistently higher in the high stocking density groups (HSD^50th^ and HSD^75th^) compared with those housed at a lower stocking density (LSD^50th^ and LSD^75th^). The welfare indicator “bedding quantity” was also judged as inadequate in both HSD^50th^ and HSD^75th^. Indeed, when the lower cut-off value was set to 3.95 m^2^/horse (i.e., the 50th percentile), the frequency of pens in the HSD^50th^ group scored as containing an inadequate quantity of bedding was 83.3%, whereas when the cut-off value was set to 4.75 m^2^/horse (i.e., the 75th percentile), the frequency of pens scored as having an inadequate quantity of bedding was 78.2%. According to these results, when more animals are housed together, the frequency of inadequate quantities of bedding and the frequency of animals with dirty coats are higher. Indeed, the frequency of pens judged as having an inadequate level of bedding cleanliness exceeded 70% at all the stocking densities tested. This result may be a consequence of the high frequency of pens (>90%) containing abnormal faeces. The high prevalence of diarrhoea on the farm is probably related to the high level of starch in their diet. When the level of dietary starch exceeds the digestive capacity of a horse’s small intestine, undigested starch may reach the hindgut where it undergoes rapid fermentation. The changes that may occur in the hindgut environment as a consequence of this starch, such as a decrease in luminal pH and marked changes in the microbial population, may lead to diarrhoea and an increased risk of colic [38,39].

Taken together, the results show that the amount and cleanliness of the pens’ bedding were insufficient to provide an adequate level of environmental hygienic quality, which, in turn, influenced the cleanliness of the horses’ coats. Moreover, we might hypothesise that horses housed at a higher density were more likely to consume the straw bedding to satisfy their natural need for foraging, especially at times when hay was not available [40], and this may have exacerbated the problem.

With regard to the welfare principle of good health, a number of welfare indicators were influenced by the stocking density. The number of horses presenting clinical signs of a cough was significantly higher in the HSD^50th^ group (<3.95 m^2^/horse) compared with LSD^50th^ (≥3.95 m^2^/horse). It is well known that increasing the stocking density increases the risk for transmission of respiratory diseases in stabled horses [41]. Thus, an increase in the per horse space allowance of a pen would be expected to decrease the occurrence of this indicator. Indeed, the statistical significance disappeared when the 75th percentile cut-off was applied.

No significant differences between groups were noted for either tail condition or mane condition when the stocking density cut-off value was set to 3.95 m^2^/horse (50th percentile) since the scores of both groups revealed a high prevalence of mane and tail damage. However, differences were identified when the higher cut-off was applied, with horses allocated ≥4.75 m^2^/horse (LSD^75th^) less likely to incur mane or tail damage. A damaged tail might also be related to a major parasitic infestation that could cause excessive pruritus and lead to rubbing-induced injuries. It should be emphasized that no parasite management program was in force on this farm. Moreover, higher stocking densities may correlate with greater levels of contact made with the metal rail bars. It should also be noted that horizontal metal rail bars were in place at the feed bunks; thus, the crest of the neck was obliged to come into close contact with the metal rail bars during feed intake. Differences between groups were noted in the feeding behaviour of animals, independent of the cut-off value applied. However, the mane and tail condition differences were only noted when the higher cut-off value was applied, indicating that when animals were kept at stocking densities lower than 4.75 m^2^/horse (HSD^75th^), a higher number of animals incurred tail and mane damage, also due to the constraint of the metal rail bars at the feed bunks.

In addition to providing key information about health-related parameters, the direct observation of the animals is fundamental for gathering data on horse behaviour. This data is fundamental as it provides insight into how an animal perceives and interacts with its environment [42,43]. The affective state, as intended by Mellor in the fifth “mental” domain [24], is not considered by AWIN. The AWIN protocol assesses the emotional state through the evaluation of behaviours indirectly linked to positive emotional states only, since appropriate behaviour represents the freedom of animals to express normal behaviour–intending behaviours that are as close as possible to those performed in nature [23]. Within the welfare principle of appropriate behaviour, we observed that a higher number of animals were feeding in the more densely housed groups for both cut-off values (HSD^50th^ and HSD^75th^). However, feeding behaviour was spot sampled and may not indicate a long-term behaviour pattern. In contrast, the BCS constitutes a more direct indicator of feed intake over time [44] and reflects the consequences of feeding behaviour over the previous weeks [45]. What is important to underline is that when the stocking density cut-off was set to 3.95 m^2^/horse, we identified a higher number of animals judged as thin in the high stocking density condition. This may mean that space allowance can also influence the time dedicated to feeding. A reduction in the feeding space would be a problem if it precludes easy access to feed, which may also increase competition for resources and thus influence the daily growth rate [32].

Of the other welfare indicators describing appropriate behaviour, “resting in a standing position” also seemed to be influenced by the stocking density. The results suggest that the number of horses resting in a standing position was significantly higher in the groups characterized by a higher stocking density (HSD^50th^ and HSD^75th^), which could be a consequence of the lack of space and physical restriction [12]. Interestingly, the other behaviours included in the checklist (mutual grooming, resting in lying position, playing, sexual behaviour, aggressive behaviour, and stereotypic behaviour) were only detected at a very low frequency or absent altogether. However, the sampling method used may have influenced the results, as these behaviours may occur at much lower frequencies, meaning that the spot sampling method was not sensitive enough to detect their expression. The expression of certain behaviours could even be masked by the sampling method, as may be the case for stereotypic behaviour [46]. Other authors have speculated that the absence of certain behaviours might be a sign of a state of apathy [43]. Horses may be particularly sensitive to unfavorable environmental conditions, which could induce them to show apathy and become less reactive to environmental stimuli [42,47]. This condition could lead to the development of “depressive syndromes”, as reported by Fureix and colleagues [48]. Studies on the behavioural repertoire of horses reared for meat production are needed to investigate this possibility. In addition to revealing conditions of negative welfare status, behavioural indicators also provide a means of recognising positive affective experiences, especially when animals exercise agency [49], which, as defined by Mellor [24], is the expression of behaviours performed by animals in a voluntary way since they are involved in rewarding experiences.

It is also important to consider the feeding management strategies used in this kind of breeding farm. Unfortunately, it was not possible to quantify the exact amount of hay supplied to the animals. Consequently, it was not possible to calculate the exact forage intake/animal/day. Nonetheless, we estimated that animals received approximately 6 kg of hay per day. Since hay was only supplied twice a day, we can presume that horses spent long periods of time fasting during the day and night. In fact, we performed the welfare assessment checklist in the morning 2 h after food provision, and we can confirm that feed bunks were not sufficiently full to guarantee an adequate provision of hay until the evening meal. Moreover, horses were fed 8 kg/animal/day of a cereal-based commercial pelleted feed that was high in starch (55% as fed). It is well known that feeding horses with high amounts of starch can affect their welfare, leading to gastrointestinal and behavioural disorders [6]. Indeed, a number of equine studies state that starch consumption should be limited to not more than 2 g starch/kg bodyweight (BW) per meal [6,38,50]—equivalent to no more than 1 kg of starch/meal for a 500 kg horse or 1820 g/meal of the commercial cereal-based pelleted feed used in the present study. At this farm, the horses received 4 kg/animal/meal of the cereal-based commercial pelleted feed, corresponding to 2.2 kg of starch/animal/meal. Although it was not possible to measure the BW of the horses involved in the present study, according to the breeder, the animals belonging to the Italian heavy draft breeds and the French heavy draft breeds weighed approximately 500 and 550 kg, respectively. Therefore, we can speculate that the amount of starch fed to the animals was approximately twice the recommended safe level.

The main limitation of the present study is related to the fact that all the assessments were made in a single farm, even though it is one of the biggest meat horse breeding farms in Italy. Moreover, it was not possible to have a control group in which the minimum requirements considered by AWIN were satisfied. Even with these limitations, the present study represents the first scientific attempt to assess the welfare of horses reared for meat production at a farm level. The data obtained show the need to understand more about the welfare of those animals, stimulating further investigations to elucidate the minimum space allowance per horse in a group pen required to generate improvements in horse welfare. Measures are also needed to improve the feeding management regimes used, which should consider the nutritional requirements and welfare of the horses and not just production goals.

## 5. Conclusions

Stocking densities and feeding management regimes affect the welfare of horses reared in group pens for meat production and thus constitute key concerns. The results of the present study suggest that horse welfare is negatively affected by high stocking densities and the use of an intensive feeding management strategy. According to our results, when the horses had more than 4.75 m^2^/horse, many parameters were affected (i.e., improvement of coat cleanliness, improvement of bedding quantity, improvement of the mane and the tail condition, less resting in a standing position and less feeding related to the greater space available at the feed bunk). A further increment of space and/or changes in management regimes may be necessary to improve all the welfare indicators. The horses of this study were fed rations rich in starch, which was probably responsible for the high incidence of diarrhea and, consequently, the poor state of bedding cleanliness. This present study highlights the need for developing specific guidelines and rules for farming equines in order to safeguard their welfare. Moreover, since there is a lack of science-based minimum requirements for space allowance and feeding space for group-housed horses, this work hopes to stimulate and encourage further scientific inquiries into the management practices applied in horse farms for meat production.

## Figures and Tables

**Table 1 animals-10-01103-t001:** Area (m^2^) and feed bunk length (m) of the 12 multiple pens evaluated in the seven surveys conducted between April and June. The median values (plus 25th–75th percentiles) for the number and the height (at the withers) of the horses within each pen are reported.

Pen ID	Area of the Pen (m^2^)	Length of Feed Bunk (m)	Number of Horses Median (25th–75th)	Height at the Withers (cm) Median (25th–75th)
1	18.1	3.9	2.5 (2–3)	150 (145–150)
2	14.9	3.2	4 (4–4)	140 (137.5–140)
3	20.8	4.6	4 (4–5)	140 (140–140)
4	22.5	4.7	5 (5–6)	140 (140–143.8)
5	16.5	4.0	5 (4–5)	140 (136.3–147.5)
6	27.7	6.7	7 (7–7.75)	140 (130–140)
7	35.0	7.0	9.5 (9–10)	140 (140–150)
8	38.0	7.6	10 (9–11)	130 (130–132.5)
9	36.0	4.8	8 (7.5–8)	147.5 (141.3–153.8)
10	36.8	4.9	10 (9–11)	140 (136.3–140)
11	34.9	4.7	12 (10–13)	140 (140–145)
12	46.5	6.2	15 (14–15)	125 (125–125)

**Table 2 animals-10-01103-t002:** Welfare assessment checklist used in each of the seven surveys. The checklist is divided into four sections corresponding to the Welfare Quality^®^ principles: good feeding, good housing, good health, and appropriate behaviour. Each principle is measured using specific resource-based, management-based, and animal-based indicators. Each section is accompanied by detailed guidance notes and photographs illustrating the scores.

Welfare Principles	Welfare Criteria	Welfare Indicators	Score	Notes				
Good feeding	Appropriate nutrition	N of horses within the group pen	□ ……					
		BCS ^1^	□ N of horses scored as Thin … □ N of horses scored as Normal … □ N of horses scored as Fat …	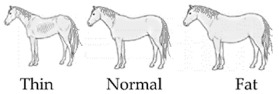				
		Length of the feed bunk	□ …… m	Consider as adequate space at the feed bunk of at least 1 m per horse (m/horse)				
		Space allowance per horse at the feed bunk (m/horse) ^2^	□ Adequate □ Inadequate					
	Absence of prolonged thirst	Water availability ^3^	□ Adequate □ Inadequate	Consider the functioning of the automatic drinkers				
		Water point cleanliness ^3^	□ Clean: Bowl and water are clean □ Partly dirty: Bowl is dirty but water is clean □ Dirty: Bowl and water are dirty	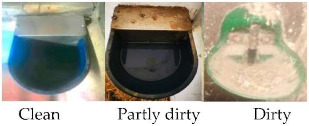				
Good housing	Comfort around resting	Bedding quantity ^4^	□ Adequate □ Inadequate	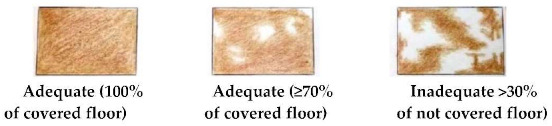				
		Bedding cleanliness ^5^	□ Adequate □ Inadequate	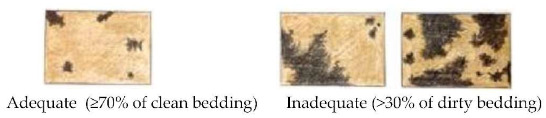				
		Coat cleanliness ^6^	□ N of horses scoring 1 … □ N of horses scoring 2 … □ N of horses scoring 3 … □ N of horses scoring 4 … □ N of horses scoring 5 …	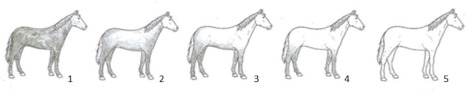				
	Thermal comfort	Environmental temperature (°C) ^7^	□ Adequate □ Inadequate	Environmental temperature is considered adequate if it ranges between +5–+25 °C				
		Environmental humidity (%) ^7^	□ Adequate □ Inadequate	Environmental humidity is considered adequate if it ranges between 60–80%				
	Ease of movement	Area of the pen (m^2^)	□ ........ m^2^	Medium height at the withers Available space per horse (m^2^/horse)	<120 cm 5.5 m^2^	120–148 cm 7 m^2^	148–162 cm 8 m^2^	162–175 cm 9 m^2^
		Medium height at the withers of the horses within the pen	□ …… cm					
		Stocking density (m^2^/horse) ^8^	□ Adequate □ Inadequate					
Good health	Absence of injuries	Integument alterations ^3^	N of horses within the pen that present integument alterations …….	Consider integument alteration: area of alopecia, skin lesions as superficial would or deep wound, tumefaction, and swelling				
		Mane condition ^9^	□ N of horses with a mane score of 1 … □ N of horses with a mane score of 2 … □ N of horses with a mane score of 3 …	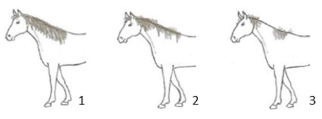				
		Tail condition ^9^	□ N of horses with a tail score of 1 … □ N of horses with a tail score of 2 … □ N of horses with a tail score of 3 …	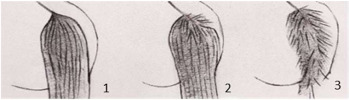				
		Swollen joints/signs of lameness ^10^	N of horses within the pen that present swollen joints/signs of lameness ……	Focus attention on distal legs, the shape of the hoof and the animals’ movements				
	Absence of diseases	Coughing ^10^	N of horses within the pen with coughing……	Evaluate coughing together with breathing assessment				
		Abnormal breathing ^10^	N of horses within the pen with abnormal breathing……	Consider breathing abnormal if the horse shows any of the following signs: flared nostrils, extended head and neck, increased respiratory rate, or asynchrony between movements of the chest and the abdomen				
		Discharges ^10^	N of horses within the pen with discharges……	Consider nasal and ocular discharges				
		Consistency of faeces ^11^	□ Normal □ Abnormal	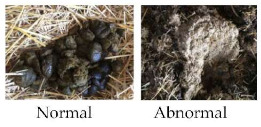				
	Absence of pain and pain induced by management procedures	State of the awareness	N of horses within the pen with an abnormal state of the awareness ………	State of awareness is considered abnormal if horses appear apathetic, depressed, alarmed, in a state of stupor				
Appropriate behaviour	Expression of social behaviour	Mutual grooming	N of horses within the pen ……	Body cleaning is performed by one horse towards a conspecific or reciprocally				
		Playing	N of horses within the pen ……	Horse plays alone or with other horses. It includes playing with structural parts of the pen, locomotor play, play fighting				
	Expression of other behaviours	Feeding	N of horses within the pen ……	Horse eats hay, straw or feedstuff in the trough or on the ground				
		Watching	N of horses within the pen ……	Horse is in a standing position. The expression is attentive, observing the surroundings				
		Resting in standing position	N of horses within the pen ……	Horse is in a standing position. The expression is relaxed				
		Resting in lying position	N of horses within the pen ……	Horse is lying on the ground in sternal position with the limbs flexed below the body or in lateral position with extended limbs				
		Sexual behaviours	N of horses within the pen ……	Stallion sniffs or bites the mare’s genitals. The stallion mounts the mare				
		Aggressive behaviours	N of horses within the pen ……	They include snaking (horse stretches its neck towards a conspecific with ears pinned back, threatening to bite); kicking (horse makes a kicking movement towards another horse with one or both hind limbs); biting (horse touches the body of another horse using its teeth whilst its ears are turned backwards).				
		Stereotypic behaviours	N of horses within the pen ……	Horse presents stereotypic behaviour: oral and/or locomotor stereotypic behaviours				

^1^ BCS was scored as thin, normal, or fat on the basis of the visual appraisal of the shape of each animal. ^2^ Space allowance at the feed bunk was considered adequate if it allowed at least 1 m per horse, as per the suggestions provided by the Code of Practice for the Care and Handling of Equines [29]. ^3^ Scores adapted from Animal Welfare Indicators (AWIN) welfare assessment protocol for horses [21]. Water availability was assessed adequate when automatic drinkers were functioning. ^4^ A specific scoring system was developed by the authors to evaluate bedding quantity. Bedding quantity was scored as adequate if ≥70% of the floor was covered by bedding. Bedding quantity was scored as inadequate if >30% of the floor was not covered by bedding. ^5^ A specific scoring system was developed by the authors to evaluate bedding cleanliness. Bedding cleanliness was scored as adequate if ≥70% of the bedding was clean, and inadequate when >30% of the bedding was dirty. ^6^ Specific 5-point scoring system developed for the assessment of coat cleanliness: 1: coat completely dirty; 2: dirty limbs, abdomen, barrel, flanks, and neck; 3: dirty limbs, and abdomen; 4: dirty limbs; 5: coat completely clean. ^7^ Scores adapted from Wageningen UR Livestock Research Welfare Monitoring System Assessment protocol for horses [30]. Temperature was considered adequate when it was within the horse’s thermoneutral zone (+5 to +25 °C). Relative humidity was deemed to be adequate when the values ranged from 60 to 80%. ^8^ Stocking density was considered adequate according to the indications reported in the associated guidance notes adapted from the AWIN protocol in the section for group-housed horses [21] (i.e., if horses are assessed to measure between 120 and 148 cm at the withers, a minimal space of 7 m^2^/horse is required to be considered adequate). ^9^ Specific 3-point scoring system defining mane and tail condition: 1: mane/tail are in good condition for their entire length; 2: areas of broken and/or absent mane or tail hair, but intact skin; 3: areas of broken and/or absent mane or tail hair and damaged skin. ^10^ Scores adapted from AWIN welfare assessment protocol for horses [21]. ^11^ Faeces were scored as normal if the shape of the faeces was conserved.

**Table 3 animals-10-01103-t003:** Statistical analysis performed using the median cut-off value for the stocking density (3.95 m^2^/horse). Nondichotomous variables are expressed as the median number of horses (plus 25th–75th percentiles) within pens that show a specific score or health condition or are performing a specific behaviour. Space at the feed bunk is expressed as the median (plus 25th–75th percentiles) length in metres available per horse. Nondichotomous variables were analysed using the Mann–Whitney U test: the test statistic (U) and *p*-values are reported. Dichotomous variables are expressed as frequencies (%) and were analysed using the Fisher exact test: the test statistic (χ^2^) and *p*-values are reported. Data were considered significant for *p*-values < 0.05.

Welfare Principle	Welfare Indicator	LSD^50th^ Median Values (25th–75th Percentiles) and Frequencies (%) for Groups (*n* = 37) with ≥3.95 m^2^/Horse	HSD^50th^ Median Values (25th–75th Percentiles) and Frequencies (%) for Groups (*n* = 36) with <3.95 m^2^/Horse	Test Statistics ^§^ Mann–Whitney U Test (U) Fisher Exact Test (χ^2^)	*p*-Values
Good feeding	Space at feed bunk (m/horse)	0.95 (0.70–1.30)	0.6 (0.42–0.79)	U = 194.00	<0.001 *
	BCS ^0^	0 (0–0)	0.5 (0–2.25)	U = 459.00	0.004 *
	Water point cleanliness ^a^	Adequate: 68.6% Inadequate: 31.4%	Adequate: 63.9% Inadequate: 36.1%	χ^2^ = 0.174	0.803
Good housing	Coat cleanliness ^1^	3 (1–4)	5 (2–7)	U = 408.50	0.004 *
	Bedding cleanliness ^a^	Adequate: 22.9% Inadequate: 77.1%	Adequate: 16.7% Inadequate: 83.3%	χ^2^ = 0.387	0.757
	Bedding quantity ^a^	Adequate: 43.2% Inadequate: 56.8%	Adequate: 16.7% Inadequate: 83.3%	χ^2^ = 6.121	0.021 *
Good health	Skin lesions ^2^	1 (0.5–2)	1 (0–2)	U = 658.50	0.931
	Mane condition ^3^	4 (3–7)	5.5 (3–9.5)	U = 389.00	0.142
	Tail condition ^4^	1 (0–1.5)	1.5 (0–4)	U = 470.00	0.056
	Swollen joints ^5^	0 (0–1)	1 (0–2)	U = 602.00	0.444
	State of awareness ^6^	0 (0–0)	0 (0–0)	U = 610.50	0.075
	Abnormal breathing ^7^	1 (0–1)	0 (0–0.75)	U = 631.50	0.626
	Nasal discharges ^8^	0 (0–2)	0 (0–1)	U = 574.00	0.249
	Ocular discharges ^9^	0 (0–1)	0 (0–1)	U = 650.00	0.833
	Consistency of faeces ^a^	Adequate: 0% Inadequate: 100%	Adequate: 8.3% Inadequate: 91.7%	χ^2^ = 3.215	0.115
	Cough ^a^	0 (0–0)	0 (0–1)	U = 522.00	0.028 *
Appropriate behaviour	Feeding ^10^	2 (0.5–4)	5 (2–6.75)	U = 353.50	0.001 *
	Watching ^11^	1 (0–3)	1.5 (0–4)	U = 598.00	0.442
	Mutual grooming ^12^	0 (0–0)	0 (0–0)	U = 648.50	0.574
	Resting in a standing position ^13^	0 (0–2)	1 (0–3)	U = 452.00	0.012 *
	Resting in a lying position ^14^	0 (0–0)	0 (0–1)	U = 597.50	0.306
	Playing ^15^	0 (0–0)	0 (0–0)	U = 623.00	0.574
	Sexual behavior ^16^	0 (0–0)	0 (0–0)	U = 646.50	0.532
	Aggressive behavior ^17^	0 (0–0)	0 (0–0)	U = 566.00	0.076
	Stereotypic behavior ^18^	0 (0–0)	0 (0–0)	U = 666.00	1

* Significant values. ^§^ The degrees of freedom for each analysed variable were equal to 1. ^a^ Dichotomous variables expressed as frequencies (%) of occurrence within the multiple pens. ^0^ N of horse scored as thin using the specifically developed 3-point scoring system. ^1^ N of horses with a coat cleanliness score of 1, 2 or 3, using the specifically developed 5-point scoring system. ^2^ N of horses within the pens presenting skin lesions, including areas of alopecia, injuries, tumefaction, or swelling. ^3^ N of horses presenting a ruined mane, as defined by a score of 3, using the specifically developed 3-point scoring system. ^4^ N of horses presenting a ruined tail, as defined by a score of 3, using the specifically developed 3-point scoring system. ^5^ N of horses presenting swollen joints. ^6^ N of horses presenting an abnormal state of awareness. ^7^ N of horses presenting abnormal breathing. ^8^ N of horses presenting nasal discharges. ^9^ N of horses presenting ocular discharges. ^10^ N of horses feeding. ^11^ N of horses watching. ^12^ N of horses engaged in mutual grooming. ^13^ N of horses resting in a standing position. ^14^ N of horses resting in a lying position. ^15^ N of horses playing. ^16^ N of horses performing sexual behaviours. ^17^ N of horses engaged in aggressive behaviours. ^18^ N of horses performing stereotypic behaviours.

**Table 4 animals-10-01103-t004:** Statistical analysis performed using the 75th percentile cut-off value (4.75 m^2^/horse). Nondichotomous variables are expressed as the median number of horses (plus 25th–75th percentiles) presenting a specific score or health condition or performing a specific behaviour. Space at the feed bunk is expressed as median (plus 25th–75th percentiles) length in metres available per horse. Nondichotomous variables were analysed using the Mann–Whitney U test: test statistic (U) and *p*-values are reported. Dichotomous variables are expressed as frequencies (%) and were analysed using the Fisher exact test: the test statistic (χ^2^), degrees of freedom and *p*-values are reported. Data were considered significant for *p*-values < 0.05.

Welfare Principle	Welfare Indicator	LSD^75th^ Median Values (25th–75th Percentiles) and Frequencies (%) for Groups (*n* = 18) with ≥4.75 m^2^/Horse	HSD^75th^ Median Values (25th–75th Percentiles) and Frequencies (%) for groups (*n* = 55) with <4.75 m^2^/Horse	Test Statistics ^§^ Mann–Whitney U test (U) Fisher Exact Test (χ^2^)	*p*-Values
Good feeding	Space at feed bunk (m/horse)	1.3 (1.10–1.54)	0.70 (0.45–0.84)	U = 95.00	<0.001 *
	BCS ^0^	0 (0–0)	0 (0–2)	U = 388.00	0.105
	Water point cleanliness ^a^	Clean (0): 62.5% Dirty (1): 37.5%	Clean (0): 67.3% Dirty (1): 32.7%	χ^2^ = 0.126 (1)	0.769
Good housing	Coat cleanliness ^1^	2 (1–4)	4 (2–7)	U = 275.50	0.005 *
	Bedding cleanliness ^a^	Adequate: 29.4% Inadequate: 70.61%	Adequate: 16.7% Inadequate: 83.3%	χ^2^ = 1.275 (1)	0.299
	Bedding quantity ^a^	Adequate: 55.6% Inadequate: 44.4%	Adequate: 21.8% Inadequate: 78.2%	χ^2^ = 7.331 (1)	0.016 *
Good health	Skin lesions ^2^	1 (0–2)	1 (0–2)	U = 443.50	0.49
	Mane condition ^3^	3.5 (3–4.75)	5 (3–8)	U = 245.50	0.038 *
	Tail condition ^4^	0 (0–1)	1 (0–3)	U = 313.50	0.024 *
	Swollen joints ^5^	0 (0–1)	1 (0–2)	U = 374.50	0.095
	State of awareness ^6^	0 (0–0)	0 (0–0)	U = 468.00	0.315
	Abnormal breathing ^7^	0 (0–1)	0 (0–1)	U = 494.00	0.095
	Nasal discharges ^8^	0 (0–2)	0 (0–1)	U = 484.00	0.873
	Ocular discharges ^9^	0 (0–0)	0 (0–1)	U = 391.00	0.113
	Consistency of faeces ^a^	Adequate: 0% Inadequate: 100%	Adequate: 5.5% Inadequate: 94.5%	χ^2^ = 1.024 (1)	0.570
	Cough ^a^	0 (0–0)	0 (0–0.5)	U = 420.00	0.183
Appropriate behaviour	Feeding ^10^	1.5 (0–3.25)	3 (2–6)	U = 260.50	0.002 *
	Watching ^11^	1.5 (0–2.25)	1 (0–4)	U = 413.00	0.282
	Mutual grooming ^12^	0 (0–0)	0 (0–0)	U = 449.00	0.087
	Resting in a standing position ^13^	0 (0–0.25)	1 (0–3)	U = 277.50	0.003 *
	Resting in a lying position ^14^	0 (0–0.25)	0 (0–0)	U = 489.00	0.917
	Playing ^15^	0 (0–0)	0 (0–0)	U = 418.00	0.125
	Sexual behavior ^16^	0 (0–0)	0 (0–0)	U = 468.00	0.315
	Aggressive behavior ^17^	0 (0–0)	0 (0–0)	U = 396.00	0.120
	Stereotypic behavior ^18^	0 (0–0)	0 (0–0)	U = 495.00	1

* Significant values. ^§^ The degrees of freedoms for each analysed variable were equal to 1. ^a^ Dichotomous variables expressed as frequencies (%) of occurrence within the multiple pens. ^0^ N of horses scored as thin using the specifically developed 3-point scoring system. ^1^ N of horses with a coat cleanliness score of 1, 2 or 3, using the specifically developed 5-point scoring system. ^2^ N of horses within the pens presenting skin lesions, including areas of alopecia, injuries, tumefaction, or swelling. ^3^ N of horses presenting a ruined mane, as defined by a score of 3, using the specifically developed 3-point scoring system. ^4^ N of horses presenting a ruined tail, as defined by a score of 3, using the specifically developed 3-point scoring system. ^5^ N of horses presenting swollen joints. ^6^ N of horses presenting an abnormal state of awareness. ^7^ N of horses presenting abnormal breathing. ^8^ N of horses presenting nasal discharges. ^9^ N of horses presenting ocular discharges. ^10^ N of horses feeding. ^11^ N of horses watching. ^12^ N of horses engaged in mutual grooming. ^13^ N of horses resting in a standing position. ^14^ N of horses resting in a lying position. ^15^ N of horses playing. ^16^ N of horses performing sexual behaviours. ^17^ N of horses engaged in aggressive behaviours. ^18^ N of horses performing stereotypic behaviours.

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
