# Peer review of "Stocking Density Affects Welfare Indicators in Horses Reared for Meat Production"

_animals, 2020, doi:10.3390/ani10061103_

Round 1
Reviewer 1 Report
I found the results of the paper quite interesting and I think this is an under-explored topic which deserves some experimental attention. There are some aspects of the paper which should be addressed before publication.
The introduction is exceedingly brief and should be expanded. There is a lot of relevant information that could be provided to the reader so that they understand the context and to ‘join the dots’ between the statements that are made. For example, what are some challenges for horse welfare? Why is there ‘consequently some concern for animal welfare’ if animals are kept in intensive farming systems? Why is it important to know about the horse statistics in Italy compared to other European countries that also eat horse? What legislation covers horse welfare? What horses are used for meat (e.g. breeds, age). What is the Animal Welfare Indicators? What is Welfare Quality? What are the Welfare Quality principles? One of the major issues is that the authors assume the reader already has a lot of the background knowledge, but the introduction should actually provide this. The introduction should pave the way for the aims, so that we understand exactly why this study is important to conduct, and why the researchers had specific hypotheses.
There are numerous language errors and the writing should be carefully edited to improve clarity of the information, and general flow of the writing. It might help to ask someone who is highly proficient in English scientific writing but is not on the research team so that they can use “fresh eyes” to look over the manuscript.
For the methods, I suggest to provide a table in the methods section showing the groups, when they were observed, their demographic details, their pen size etc, so that the reader can understand more clearly how the study was carried out before the results section.
The paper suggests that the AWIN protocol for horses was used, however the welfare checklist has clear differences. For example, in AWIN body condition score is on a scale of 1-5 while in the paper three categories (thin, normal, fat) was used. AWIN also contains lots of important welfare indicators such as hoof health, and these have been dropped out. Clarification should be given around what was used directly from other validated assessments, and what needed to be adjusted and the reason. In essence, it is not clear why the authors have deviated from validated welfare assessments, and more detail needs to be given on why available assessments aren’t suitable in this context, and how the re-developed one addresses this problem.
It isn’t terribly clear what the point of the study is. Is it to assess welfare within meat horses? Is it to develop a tool that can be used in this context? Is it to assess whether AWIN is suitable for this context? Is it to assess stocking density? Once the aims section of the paper is restructured and clarified it will be easier to ensure that the rest of the paper is structured to address the aims.
Specific comments:
- It is better to start each sentence with the main information. Here, a much stronger start would be to start with “More than 500 000 horses are slaughtered each year in Europe for meat production”. The information about sporting or pet horses is irrelevant and detracts from the most important information. This is an issue throughout but another example to help you identify this is at L56 where the sentence would be easier to read if it was rearranged “This work also aimed to evaluate if welfare was affected by stocking density within pens. We hypothesised that welfare would be poorer as stocking density increased.”
- Put the aims and hypotheses into their own paragraph.
- This seems like an odd aim. Wouldn’t the aim be to assess meat horse welfare? Why is it a specific part of the aim to use AWIN?
- It isn’t clear whether ethical approval was applied for, or needed? Please add a specific statement about any institutional ethics approvals that were granted, or if the study was exempt from needing this.
- The details given here may be better after the data collection section so that the reader understands the study site first.
- “was” change to “were”
- The term “invasive” can have several meanings and should therefore be defined
- Biggest farm of? Horse farm? Biggest farm in general? Clarify
- “In” should be “on” and “over” might be better as “with”
- Change to “summer of 2017”
- Was the survey at 9:30am or the feeding? Clarify
- Were the pens used chosen randomly?
- “both sexes” rather than “different sexes”
- L70 Were they in a shed-like building? Or did it have open sides?
- Provide the feed name and company
- Take out “the” before “50%”
- Unsure what is meant by “that fulfilled its own checklist separately”
- Provide more detail on the expert evaluators – what is their background? How familiar are they with the welfare assessment? How familiar are they with horses? Was any reliability testing carried out prior to collecting the study data?
- Are the principles being evaluated, or the horse welfare being evaluated? Perhaps re-frame the sentence
- Change “be” to “include” or change “and” to “or”
- Please note that I stop correcting small edits here but the authors should make a thorough assessment of all of the paper to improve these subtle but numerous errors
- What is meant by “adapted to the specific breeding condition”?
- Quite a few of the methods seem to be more blunt than in the AWIN. For example the number of scored categories is larger in AWIN for body condition score and water cleanliness than in this paper. Unclear what the reason for this is and whether it affects the precision of the scoring.
- Table 1. “State of the sensory” is not a clear term because sensory is an adjective, not a noun
- Table 1 & L165. Behaviour like “coughing” is included but it is not clear what the protocol is for assessing these intermittant behaviours, unlike in AWIN where data collection protocols are clearly outlined (e.g. assess for a 5 minute period). The same is true for the other behaviours like mutual grooming. What kind of behaviour sampling and recording is being used? At L179 it suggests spot sampling which may mean scan sampling? But no details are given on the scanning interval, whether all animals are scannd, whether they are done methodically (e.g. scan left to right) and how many scans were performed.
- Table 1. Reducing aspects like temperature to “adequate” and “inadequate” firstly drops out a lot of detail in terms of where the environment sits within this range, and secondly provides no information on which side of adequate an inadequate score is (e.g. is it too hot or too cold?).
- Why use a Canadian code of practice for an Italian study?
- “Bowel” should be “bowl”
- Why is coat cleanliness included rather than coat condition as per AWIN?
- Is stocking density measured in m2/horse? From the description here and in AWIN it seems that it is a measure of m2 per cm wither height of the horses. Also, how was wither height measured accurately as the horses weren’t handled?
- From “The stocking density”, it would be better if the reader didn’t have to continually refer back to AWIN. Make the methods self-contained as much as possible.
- What strategies were used to minimise observer bias and observer effects?
- L194 & 196: Correct spelling of “Dichotomous”. Also – some of these aren’t dichotomous (e.g. BCS is a three-point scale – although it looks like good and fat were combined but I couldn’t see explanation of that. Apologise if I have missed it?).
- Open bracket after age needs a closing bracket
- “Pen” should be “pens”
- Put “with” in the brackets before 25th
- In the results, the full test details need to be reported. The exact details are dependent on the test used but should include test statistic and may include degrees of freedom. It isn’t enough to just include the p value.
- The terms wide and narrow stocking density should be reconsidered because they don’t reflect the conditions well, because they imply a certain pen shape. High and low stocking density would be more usual I believe
- Table 3 – how can hair coat cleanliness have a 25-75th percentile of 7 for the NSD group when the cleanliness score only went up to 5. Check the other measures for similar issues.
- It isn’t made clear in the study why one set of analyses use a median cut off and the second set uses the 75th percentile. This requires more explanation and justification.
- When conducting the statistical analyses was the lack of independence between horses accounted for in any way? When animals are grouped, it is usually the group that is the experimental unit to avoid pseudoreplication. How has this been addressed in the study.
- In the discussion, the results need to be discussed more in light of current literature. Very little welfare literature has been included or discussed in this paper which makes it difficult to understand how it fits within the current body of scientific literature. This aspect needs to be considerably improved.
Author Response
Dear Reviewer,
We have answered every concern point by point. The answers are visible in red.
We attach the file where the changes are visible in red. The manuscript was revised by a professional English service. Only the English language of the answers was not reviewed by the professional English service.
Thank you for your precious work done to improve the quality of our manuscript.
Reviewer 1
I found the results of the paper quite interesting and I think this is an under-explored topic which deserves some experimental attention. There are some aspects of the paper which should be addressed before publication.
Thank you for your comment and for appreciating our efforts. Thank you also for your precious help to improve the quality of our manuscript. We did our best efforts to follow your suggestions.
The introduction is exceedingly brief and should be expanded. There is a lot of relevant information that could be provided to the reader so that they understand the context and to ‘join the dots’ between the statements that are made. For example, what are some challenges for horse welfare? Why is there ‘consequently some concern for animal welfare’ if animals are kept in intensive farming systems? Why is it important to know about the horse statistics in Italy compared to other European countries that also eat horse? What legislation covers horse welfare? What horses are used for meat (e.g. breeds, age). What is the Animal Welfare Indicators? What is Welfare Quality? What are the Welfare Quality principles? One of the major issues is that the authors assume the reader already has a lot of the background knowledge, but the introduction should actually provide this. The introduction should pave the way for the aims, so that we understand exactly why this study is important to conduct, and why the researchers had specific hypotheses.
Thank you for your comment. Now Introduction is rewritten following your suggestions and the suggestions of the referee 2.
There are numerous language errors and the writing should be carefully edited to improve clarity of the information, and general flow of the writing. It might help to ask someone who is highly proficient in English scientific writing but is not on the research team so that they can use “fresh eyes” to look over the manuscript.
Thank you for your suggestion. The revised version is now edited by a professional English service.
For the methods, I suggest to provide a table in the methods section showing the groups, when they were observed, their demographic details, their pen size etc, so that the reader can understand more clearly how the study was carried out before the results section.
Thank you for this suggestion. Table (as Table 1) is now shown in the method section (Line 127).
The paper suggests that the AWIN protocol for horses was used, however the welfare checklist has clear differences. For example, in AWIN body condition score is on a scale of 1-5 while in the paper three categories (thin, normal, fat) was used. AWIN also contains lots of important welfare indicators such as hoof health, and these have been dropped out. Clarification should be given around what was used directly from other validated assessments, and what needed to be adjusted and the reason. In essence, it is not clear why the authors have deviated from validated welfare assessments, and more detail needs to be given on why available assessments aren’t suitable in this context, and how the re-developed one addresses this problem.
It isn’t terribly clear what the point of the study is. Is it to assess welfare within meat horses? Is it to develop a tool that can be used in this context? Is it to assess whether AWIN is suitable for this context? Is it to assess stocking density? Once the aims section of the paper is restructured and clarified it will be easier to ensure that the rest of the paper is structured to address the aims.
Thank you for your comments. We have underlined in the introduction that we used an adapted version of AWIN protocol. We are totally agreeing with you that 4 principles and criteria are validated. However, when we decided to apply directly the AWIN protocol we had some problems, because of the breeding conditions in which horses reared for meat production are kept. In fact, the AWIN protocol is a great tool but it is specifically developed for single stabled horse or grouped housed horses. As you know AWIN protocol has some limitations about its application on horses reared for meat production. In particular, in the same AWIN protocol it is expressed the need to refine and improve the section of group housed horses in light of the progress of scientific research. Moreover, AWIN protocol is intend for horses with more than 5 years old. As a consequence, this tool seems difficult to apply in intensive breeding farms that rear young horses kept in group pens. For example, it is not possible to judge the BCS directly on the animals since they are kept in group pens and it is very difficult to touch the animals. For this reason, we have decided to split the BCS in 3 categories, as described in the methods section.
Now, we have rewritten the introduction and the aims, adding also comments of reviewer 2 that asked us to add feeding management in aims section (Lines 88-96). Moreover, methods are rewritten in light of your and reviewer 2’ suggestions.
Specific comments:
L42 It is better to start each sentence with the main information. Here, a much stronger start would be to start with “More than 500 000 horses are slaughtered each year in Europe for meat production”. The information about sporting or pet horses is irrelevant and detracts from the most important information. This is an issue throughout but another example to help you identify this is at L56 where the sentence would be easier to read if it was rearranged “This work also aimed to evaluate if welfare was affected by stocking density within pens. We hypothesised that welfare would be poorer as stocking density increased.”
Thank you for your suggestions. Now Introduction is rewritten (Lines 38-96) thanks to your and reviewer 2 suggestions. We have done most efforts over all the manuscript to meet your comments and English is now revised.
L54 Put the aims and hypotheses into their own paragraph.
Done (Lines 88-96).
L54 This seems like an odd aim. Wouldn’t the aim be to assess meat horse welfare? Why is it a specific part of the aim to use AWIN?
Now aims are rewritten (Lines 88-96). We used the AWIN protocol since it is the only tool validated for horses by the European Commission. For this reason, we thought it was a starting point for our study. However, since the breeding conditions in which horses reared for meat production are kept we made some adaptations and refinements to the protocol. These modifications have been done without changing the main structure of the protocol that is based on welfare principles and welfare criteria that are well recognised by the scientific community. The aim of the study was to evaluate if the stocking density and the feeding management affected the welfare indicators that we have selected and adapted on the basis of the farming conditions in meat horse breeding farm. Moreover, we tested the effect of two different stocking density cut-off values to assess whether any improvements in horse welfare could be observed with even a small increase in space allowance per horse.
L60 It isn’t clear whether ethical approval was applied for, or needed? Please add a specific statement about any institutional ethics approvals that were granted, or if the study was exempt from needing this.
Specific sentence is now added (Lines 98-101)
L60 The details given here may be better after the data collection section so that the reader understands the study site first.
Thank you for your comment. We have moved this information in data collection (Line 107).
L61 “was” change to “were”
Now sentence is rewritten (Lines 98-101).
L61 The term “invasive” can have several meanings and should therefore be defined
Sentence is now rewritten (Lines 98-101).
L64 Biggest farm of? Horse farm? Biggest farm in general? Clarify
Corrected (Line 103).
L64 “In” should be “on” and “over” might be better as “with”
Sentence is now rewritten (Lines 104-105).
L65 Change to “summer of 2017”
Sentence is now rewritten (Line 104).
L65 Was the survey at 9:30am or the feeding? Clarify
Sentence is now reworded (Line 105).
Were the pens used chosen randomly?
Thank you for your question. Sentence is now reworded to clarify the method of pens selection (Lines 12-122).
L68 “both sexes” rather than “different sexes”
Corrected (Line 107).
L70 Were they in a shed-like building? Or did it have open sides?
Sentence is now added (Line 109).
L76 Provide the feed name and company
Feed analysis of the concentrate is given in the text (Lines 118-119). We have not provided commercial name of the feedstuff but we have added the type of feed according to the 767 legislation in EU country (complementary feed) (Line 118).
L78 Take out “the” before “50%”
Sentence is now reworded to clarify the method of pens selection as suggested by all the reviewers (Lines 120-122)
L83 Unsure what is meant by “that fulfilled its own checklist separately”
Sentence is now reworded and corrected for English language, following also the suggestion of reviewer 3 (Lines 138).
L83 Provide more detail on the expert evaluators – what is their background? How familiar are they with the welfare assessment? How familiar are they with horses?
Done (Lines 133).
Was any reliability testing carried out prior to collecting the study data?
Thank you for your comment. The checklist used is based on welfare indicators that are validated. However, as described before, authors needed to add and/or modify the evaluation of some indicators (i.e. bcs, coat cleanliness …) because of the specific breeding conditions in which horses reared for meat production are kept. Before starting the study, evaluators were trained on the checklist and at the end of the training period, reliability was evaluated. We have provided to explain better this in the test (Lines 133-135).
L86 Are the principles being evaluated, or the horse welfare being evaluated? Perhaps re-frame the sentence
Thank you for the suggestion. Sentence is reworded (Lines 139-140).
L87 Change “be” to “include” or change “and” to “or”
Done (Lines 140-141).
Please note that I stop correcting small edits here but the authors should make a thorough assessment of all of the paper to improve these subtle but numerous errors
Thank you so much for your valuable comments to help us to improve our work. Now all the manuscript is revised for English language, so grammar mistakes should be corrected.
L88 What is meant by “adapted to the specific breeding condition”?
Sentence is now deleted (Line 143).
Quite a few of the methods seem to be more blunt than in the AWIN. For example the number of scored categories is larger in AWIN for body condition score and water cleanliness than in this paper. Unclear what the reason for this is and whether it affects the precision of the scoring.
Thank you for your comment. As you have suggested we have provided to give more explanations in the method section. We hope that now explanations why some parameters were scored in different way is more clear. (BCS= Lines 164-166) (Water cleanliness= Lines 175-180)
Table 1. “State of the sensory” is not a clear term because sensory is an adjective, not a noun
Corrected in all the manuscript as “state of awareness”, following the suggestion of the reviewer 3 (Table 2 – Line 143).
Table 1 & L165. Behaviour like “coughing” is included but it is not clear what the protocol is for assessing these intermittant behaviours, unlike in AWIN where data collection protocols are clearly outlined (e.g. assess for a 5 minute period). The same is true for the other behaviours like mutual grooming. What kind of behaviour sampling and recording is being used? At L179 it suggests spot sampling which may mean scan sampling? But no details are given on the scanning interval, whether all animals are scannd, whether they are done methodically (e.g. scan left to right) and how many scans were performed.
It was not possible to assess each horse for coughing for a period of 5 minutes; but evaluation was done for the time necessary to evaluate horse breathing (Table 2-Line 143, Lines 255-260).
We have provided more descriptions on the methods used in the appropriate behaviour section (Lines 277-289) following also the suggestions of the reviewer 2.
Table 1. Reducing aspects like temperature to “adequate” and “inadequate” firstly drops out a lot of detail in terms of where the environment sits within this range, and secondly provides no information on which side of adequate an inadequate score is (e.g. is it too hot or too cold?).
We have used the indication provided by the Welfare Monitoring System - Assessment protocol for horses (Wageningen UR Livestock Research) that uses this range of temperature and relatively humidity. This section is rewritten taking also into account comments of reviewer 3 (Lines 205-212).
L114 Why use a Canadian code of practice for an Italian study?
This section is rewritten (Lines 168-172). However, we have taken into account in the manuscript this document, since it is the only reference we have found about indication for recommended feeding space per horse at feed bunk when horses are housed in group.
L119 “Bowel” should be “bowl”
Corrected (Lines 176 and 178).
L134 Why is coat cleanliness included rather than coat condition as per AWIN?
Thank you for this question. We have decided to introduce the coat cleanliness that is not considered by the AWIN, since this is a problem related to the intensive meat horse breeding farm and it is not usually a problem for saddle or leisure horses. In our opinion, the coat cleanliness is an important parameter that reflect many aspects of the welfare and moreover can mask some minor coat conditions. Sentence is added in the manuscript (Lines 197-199)
L144 Is stocking density measured in m2/horse? From the description here and in AWIN it seems that it is a measure of m2 per cm wither height of the horses. Also, how was wither height measured accurately as the horses weren’t handled?
We have added sentence in the text (Lines 220-223).
L146 From “The stocking density”, it would be better if the reader didn’t have to continually refer back to AWIN. Make the methods self-contained as much as possible.
We agree with your opinion. We have provided to describe method in a more self-contained way (Lines 226-228).
L181 What strategies were used to minimise observer bias and observer effects?
We have used 5 different trained evaluators that were placed in different position in front of the pen during the assessment, and for the evaluation of the behaviours we have let the animals to accustom to the observers (Lines 283-286).
L194 & 196: Correct spelling of “Dichotomous”. Also – some of these aren’t dichotomous (e.g. BCS is a three-point scale – although it looks like good and fat were combined but I couldn’t see explanation of that. Apologise if I have missed it?).
Spelling is now corrected (Line 306). Thank you, you are right. We did mistake. We have provided to correct it (Lines 306-307).
L200 Open bracket after age needs a closing bracket
Sentence is now rewritten following also the suggestion of the reviewer 2 (Lines 311-312).
L201 “Pen” should be “pens”
Sentence is now rewritten (Lines 311-312).
L202 Put “with” in the brackets before 25th
As suggested by the English revision service we put “plus” (Line 313). Please, let us know if you agree with this correction.
In the results, the full test details need to be reported. The exact details are dependent on the test used but should include test statistic and may include degrees of freedom. It isn’t enough to just include the p value.
According to your suggestions, we have included in the Tables 3 (Line 348) and Table 4 (Line 407) all the detailed information in relation to statistical analyses used.
The terms wide and narrow stocking density should be reconsidered because they don’t reflect the conditions well, because they imply a certain pen shape. High and low stocking density would be more usual I believe
Corrected as you and referee 2 have suggested. Thank you very much.
Table 3 – how can hair coat cleanliness have a 25-75th percentile of 7 for the NSD group when the cleanliness score only went up to 5. Check the other measures for similar issues.
5 (2-7) is not referred to the score but to the median (25th-75th percentiles) number of horses scored within the pens with high stocking density (HSD50th). Descriptions are reported below the table (Line 357). Moreover, for more clarity, we have provided to improve the table text description (Lines 348-349).
It isn’t made clear in the study why one set of analyses use a median cut off and the second set uses the 75th percentile. This requires more explanation and justification.
Add in aims: We tested the effect of two different stocking density cut-off values (the median and 75th percentile values), used to divide the data into two groups (low vs. high stocking densities), to assess whether any improvements in horse welfare could be observed with even a small increase in space allowance per horse (Lines 93-96).
When conducting the statistical analyses was the lack of independence between horses accounted for in any way? When animals are grouped, it is usually the group that is the experimental unit to avoid pseudoreplication. How has this been addressed in the study.
The independence between horses was insured by the fact that even if the same 12 pens were assessed on the 7 surveys, on each occasion the horses in the pens had changed as had the stocking densities. Accordingly, the same horse was not evaluated two times. Sentence is added in the text (Lines 122-124)
In the discussion, the results need to be discussed more in light of current literature. Very little welfare literature has been included or discussed in this paper which makes it difficult to understand how it fits within the current body of scientific literature. This aspect needs to be considerably improved.
Thank you for your comment. Now discussion is rewritten and improved on this aspect, following also the suggestion of the reviewer 2.

Reviewer 2 Report
This is a really interesting paper which I would be really pleased to see published in Animals. However there is still quite alot of work to be done. The language requires amendment throughout - I understand that this is a case of non-english as first language and that is absolutely fine. I have therefore included a substantial amount of support through making language and phrasing related changes throughout the manuscript. These can be found on the attached file. There are 89 comments in total (with a few additional un-numbered ones too).
General comments that are NOT included on the annotated manuscript are:
- The introduction whilst focussed and interesting - is too short. It needs to be at least twice as long and provide more background to both welfare assessment and its importance. Also perhaps some text on horse meat as a valuable commodity, and more on legislation that fails to cover horses.
- There is no explicit reference to ethical approval for the conduct of the study (even though it was non-invasive as such).
- The method section needs to include more on the structuring of when the surveys took place in general and also the 'spot' sampling method used (do you mean scan-sampling - ref Martin and Bateson's Measuring Animal Behaviour book). Don't forget that the readers must be able to read your method and then immediately be able to implement it to replicate your study!
- A general point regarding relaying of results - rather than just saying there was a difference - specify the direction of the difference in the text.
- The discussion was enjoyable and useful but could have been extended in places - no reference to the Five Domains (this could go in the revised intro instead though). The discussion would also have benefited from a few more supporting references throughout.
- The conclusion again was useful but needed to be just one paragraph to link clearly to the title of the paper, and remove the final part which talks about future work. That should be at the end of the discussion section.
I look forward to seeing this valuable work in press in the future. Thank you
Author Response
Dear Reviewer,
We have answered every concern point by point. The answers are visible in red.
We attach the file where the changes are visible in red. The manuscript was revised by a professional English service. Only the English language of the answers was not reviewed by the professional English service.
Thank you for your precious work!
Reviewer 2
This is a really interesting paper which I would be really pleased to see published in Animals. However there is still quite alot of work to be done. The language requires amendment throughout - I understand that this is a case of non-english as first language and that is absolutely fine. I have therefore included a substantial amount of support through making language and phrasing related changes throughout the manuscript. These can be found on the attached file. There are 89 comments in total (with a few additional un-numbered ones too).
We really would like to thank you for your comment. Thank you also for your efforts to help us to improve our work.
General comments that are NOT included on the annotated manuscript are:
The introduction whilst focussed and interesting - is too short. It needs to be at least twice as long and provide more background to both welfare assessment and its importance. Also perhaps some text on horse meat as a valuable commodity, and more on legislation that fails to cover horses.
Thank you for your comment. Now introduction is rewritten, in light also of what suggested by the reviewer 1.
There is no explicit reference to ethical approval for the conduct of the study (even though it was non-invasive as such).
Specific sentence is now added (Lines 98-101).
The method section needs to include more on the structuring of when the surveys took place in general and also the 'spot' sampling method used (do you mean scan-sampling - ref Martin and Bateson's Measuring Animal Behaviour book). Don't forget that the readers must be able to read your method and then immediately be able to implement it to replicate your study!
Thank you for your suggestion. Now method section is rewritten and implemented to explain in a clear way the methods that we used. We had followed also the suggestions proposed by the reviewer 1. We hope that now methods used are more clear.
A general point regarding relaying of results - rather than just saying there was a difference - specify the direction of the difference in the text.
Thank you for your comment. We have rewritten results following your suggestion (section 3.1 – Line 317; Section 3.2 – Line 375).
The discussion was enjoyable and useful but could have been extended in places - no reference to the Five Domains (this could go in the revised intro instead though). The discussion would also have benefited from a few more supporting references throughout.
Thank you. We have provided to improve introduction and discussion following your comments together with the comments of reviewer 1.
The conclusion again was useful but needed to be just one paragraph to link clearly to the title of the paper, and remove the final part which talks about future work. That should be at the end of the discussion section.
Thank you very much. Now conclusions are rewritten.
I look forward to seeing this valuable work in press in the future.
We really want to thank you for your valuable comments and your great effort to help us to improve our work. Not always easy… but your comments are really inspiring us to do our best!
- Line 14 Simple summary: simplify
Simple summary is now rewritten.
Simple summary:
2.Line 15. Delete S
- line 16. Add S
4.line 16. Delete” represented by”. Add “the”
5.line 17. Delete “multiple”
6.line 17. Densities
7 line 17/18 simplify this sentence
7 line 19. Delete “obtained”
8 line 19. delete “critical” and put “poor”
9 line 20. add “s” on recommendations
10 line 20. Non ripetere “recommendatinons” nella stessa frase
11 A line 23. Delete “conditions”
Thank you to have helped us to improve the quality of simple summary. Now it is rewritten aiming to simplify this section as you suggested before. Grammar mistakes are now edited by the English revision service
11 B line 23. Add in aim feeding management
Done (Line 18)
Abstract:
- line 25. Delete “multiple”
We have change “multiple” with “group” (Line 23)
13 line 26. Delete “:”
Sentence is now reworded (Lines 24-27)
14 line 29. A single farm?
Yes. Sentence is added (Line 28).
15A line 30-31 not clear
Reworded (Lines 29-30).
15B line 33. Delete “thought” and put “using”
Ok (Line 32).
15C line 34. add s
Ok (Line 33).
16 line 36. Negatively?
Added (Line 34).
17 line 36. Add feeding related
Corrected (Line 34)
18 line 38. Delete s
Ok (Line 37).
19 too many keywords? …check.
We have delated group housing (Line 37).
20 Add “,” 500,000
Reworded by English editing (Line 39).
21 line 47. Rephrase “there is a lack of…”
Sentence is reworded (Lines 51-53).
22 line 49. Change protect “safeguard” may be a better word.
Ok (Line 53).
23 line 61. Change was in “were”
Sentence is rewritten (Lines 98-101).
24 line 62. Change “with” in “in”
Ok (Line 99).
25 line 64-65. Sentence structure? Divide into 2 separate sentences.
Done (Lines 103-105).
26 line 64. Add horse meat
Sentence is reworded (Line 103)
27 line 67. Remove “an”
Done (Line 106).
28 line… comment not found.
We are sorry, we had not found the comment in the text.
29 line 68. Remove “such as”. Do you mean desexed males?
Sentence is reworded (Line 107).
30 line 71. Remove “by” and put “of”
Ok (Line 112)
31 line 71. Remove “of”
Ok (Line 112)
32 line 72. Depth … new?
33 type of straw?
34 line 73. Still need to describe in a little more detail
We have changed the text since the bedding was barley straw bedding added on a daily basis by automatic disposal (Lines 112-114).
35 line 75. You must be able to give an indication as fed individually?
Sentence is added (Line 117).
36 line 75. How fed individually in group housing?
Sentence is reworded, as suggested also by referee 3 (Lines 115-118).
37 line 77. Spelling “fiber”?
Corrected (Line 119).
38 line 87. Add either/or. You must be able to give an…
We are sorry but the comment that you wrote is cut. We have added “or” (Line 141) but probably we had not meet all your question.
38 Table 1. Change multiple in “group”
Done (Line 143 –Table 2).
39 table 1. Total N of horses in pens? So the N with poor score for eg can be calculated as a %?
Yes. Now Table 2 is modified for better comprehension (Line 143).
40 Table 1. Rephrase “state of the sensory”. General outlook perhaps
Ok. It is rephrased on the basis of the suggestion of reviewer 3 (Line 143 – Table 2).
41.Table 1. Define…
Not sure to have understood your comment. We have provided to add explicative notes in the Table 2 (Line 143).
40 line 165. Reworded as Table 1.
Done (Line 271).
41 line 179. Need to tell the reader exactly what you did here, please.
Ok, we have provided to explain better our methods (Lines 277-289).
42 line 181. Displaying the behaviour.
Added (Line 288).
42B line 183 and 185 and 197. Change multiple in “group”
Ok (Line 291). Text was edited by English revision service.
43 line 187. Wide/narrow = … seems overly complex
Changed in all the manuscript, as suggested also by reviewer 1 (Line 294).
44 line 188. Move sentence …
Sentence was reworded (Lines 300-301).
45 line 191. To do what?
Sentence is now added (Lines 302-303).
46 line 195. Useful info. Can you also include this either in methods please?
Methods have been improved adding also this information (Lines 167-168; Lines 179-180; Lines 195-196; Line 204; Lines 238-239; Lines 259-260; Lines 262-263; Lines 266-267; Lines 275-276; Lines 288-289).
47 line 197. Change doing in “performing”
Done (Line 309).
47B line 200. … of breeds?
Added (Line 311).
Line 210 Table 2. Put a summary of this info into the methods please.
Table has been moved in methods section following also the suggestion of referee 1 (Line 127 – as Table 1).
47 line 213. Delete provided by. Put “of”. Add “s”
Ok (Line 318).
line 214-215 see comment 43
Corrected.
48 line 225. Delete resulted to be and put “were”
Ok (Line 320).
49 line 227. ….
I’m so sorry. I am not able to understand what it is written in your comment.
50 line 230. Add “with”
Ok (Line 339).
51 line 231. How are you defining appropriate? Mellor et al. use “agency” … discussion
Ok, this concept is now discussed in the discussion section (Lines 532-568).
52 line 238. Change not in “non”
Ok (Line 348 and 407).
53 line 239. Delate by and put using
Ok (Lines 350,351; and Lines 409,410).
Table 3 delate “s” for Indicators
Ok (Line 348 – Table 3).
Paragraph 3.2 check 3.1
Done. Paragraph 3.1 (Line 317) and 3.2 (375) are rewritten in light of your general comment to specify the direction of the difference in the text.
54 Table 4 delate “s” for principles and indicators
Ok (Line 407 – Table 4).
Discussion:
55 line 321. change get with “obtain”
56 line 324. Put “s”
57 line 324. Change possibility of with “opportunity for”
58 line 324. Change for with “in”
58 line 325. Change for with “on”
59 line 328. Change multiple with “group”
60 line 329. Add “s”
61 line 330. Change was ranging with “ranged”
62 line 333. Add “s” purpose
63 line 334. Change specifity with “reality”
59 line 337. Change multiple in “group”
64 line 342. Change below with “less”
65 line 345. Change stay in “live”
66 line 347. Delate “an”
67 line 348. Change at with “to”
68 line 351. Change with “differences were”
69 line 351. Add stocking density. Change at with “to”
70 line 357. Change tested with “applied”
71 line 357. Change not so full with “sufficiently”
71 line 359. Change rich with “high”
73 line 363. Change in with “at”
74 line 367. Change safety with “safe”
75 line 376. Change multiple in “group”
76 line 377. Change accomplish with “satisfy”
77 line 390. Change more space with “the availability of space”
78 line 391. Delete also
79 line 392. Add related
80 line 392. Change underline with “emphasize”
81 line 393. Change in with “at”
82 line 395. Change at with “to”
81B line 410. Change impossibility with “inability”
83 line 417. Remove “in particular”
This section was in part rewritten in light to the need to improve the quality of the discussion following your and reviewer 1 suggestions. English language is totally revised by English revision service.
64 line 343. Incorrect ….. (STET)
Not sure to have understood your comment. We have provided to correct the reference (Lines 475-476).
72 line 361. Fed which horse per unit set by who?
Sentence is now reworded (Lines 579-580).
83 line 401. Good point.
Thank you for your comment.
82 line 414. … so include more discussion.
Done (Line 553-563)
Conclusions
84 line 418. 1 single paragraph
Ok (Lines 593-606).
85 line 420. Change our with “The”
Ok (Line 594).
86 line 420. Shorter to keep more focussed!!
Thank you for your comment. Conclusions are rewritten (Lines 593-606)
87 line 421. Delete “it has been shown that”
Sentence is now rewritten (Lines 595-596)
87B line 422. Change “pointed the attention” in highlighted
Ok (Line 602).
88 line 423. Change in with “at”. Yes, but shorter this sentence
Ok (Lines 600-602)
89 last paragraph: …. Of discussion not in conclusion.
Moved (Lines 588-591).

Reviewer 3 Report
The use of multiple throughout is confusing. There were several pens and each pen contained multiple horses or multiple horse pens
25 Are these breeding farms? Do they raise the foals or buy adult horses?
How were the 12 pens selected from the 24 available? Random?
Where on the farm were the temperature and humidity measurements made?
Describe metal bars at feed trough. That was not mentioned until the discussion
Discussion you are assuming that the correlation between density and poor welfare means that density is the cause.
68 were the males gelded?
69 presumably the pens were outdoors too. The horses had no access to grazing or to a pasture
75 each horse was not fed individually. Each pen was provided with an amount of pelleted feed equal to 8 kg/horse. The feed was a commercial pelleted feed (crude protein…
165 and throughout ms. State of Awareness not State of the Sensory.
71 independent of the number
Table 1 is very good. Did the evaluators actually use this as a score sheet?
Spot sampling must be described more fully. Instantaneous sampling every 5 minutes?
188 the data were not normally distributed?
221 remind reader that 1m/horse is the ideal length per horse at the feed trough
83 each of whom filled out their own checklist independently
123 ease of movement
168 incomplete sentence- breathing is the numbers
Table 3 what are units for feeding?
The discussion had no line numbers
cresty neck should be crest of the neck
Reference 22 incomplete.
Author Response
Dear Reviewer,
We have answered every concern point by point. The answers are visible in red.
We attach the file where the changes are visible in red. The manuscript was revised by a professional English service. Only the English language of the answers was not reviewed by the professional English service.
Thank you for your work!
Reviewer 3
The use of multiple throughout is confusing. There were several pens and each pen contained multiple horses or multiple horse pens
We have provided to modify the sentence “multiple pens” in “group pens”, taking into account the suggestions provided by the reviewer 2.
25 Are these breeding farms? Do they raise the foals or buy adult horses?
Sentence is added in Introduction (Lines 40-44).
How were the 12 pens selected from the 24 available? Random?
Sentence is now rewritten in light to better explain the method of pens selection used (Lines 120-124).
Where on the farm were the temperature and humidity measurements made?
Sentence is added (Lines 205-212).
Describe metal bars at feed trough. That was not mentioned until the discussion
Thank you for your comment. Sentence is now added in method section (Line 110-111)).
Discussion you are assuming that the correlation between density and poor welfare means that density is the cause.
Yes, this was the hypotheses that we tested: thus welfare would be poorer as stocking density increased. Aims are rewritten, adding also feeding management in aims as suggested by reviewer 2 (Lines 88-96) and in discussion (Lines 435-440)
68 were the males gelded?
Sentence is reworded and description is added (Line 107).
69 presumably the pens were outdoors too. The horses had no access to grazing or to a pasture
Sentence is now reworded and English language was revised by the English revision service (Line 109-110).
75 each horse was not fed individually. Each pen was provided with an amount of pelleted feed equal to 8 kg/horse. The feed was a commercial pelleted feed (crude protein…
Thank you for your suggestion. Sentence is now rewritten (Lines 115-118).
165 and throughout ms. State of Awareness not State of the Sensory.
Thank you. It is now corrected over all the manuscript as you suggest (Line 271).
71 independent of the number
Corrected (Line 112).
Table 1 is very good. Did the evaluators actually use this as a score sheet?
Thank you to appreciate our work. Yes, we did. Sentence is now added (Line 136).
Spot sampling must be described more fully. Instantaneous sampling every 5 minutes?
More explanation is now added (Lines 277-289).
188 the data were not normally distributed?
Sentence is added (Line 301).
221 remind reader that 1m/horse is the ideal length per horse at the feed trough
Thank you for your suggestion. Sentence is added (Lines 325-326).
83 each of whom filled out their own checklist independently
Sentence is corrected (Line 137).
123 ease of movement
Corrected (Line 183).
168 incomplete sentence- breathing is the numbers
Thank you, sentence is now rewritten in light also of English revision (Lines 254-256).
Table 3 what are units for feeding?
Sorry, probably we have not understood your question. I think that you refer to the number of horses that were feeding at the moment of the welfare assessment? Explanation is given below the Table 3 and in method section (Line 366).
The discussion had no line numbers
Sorry, we have provided to add line numbers
cresty neck should be crest of the neck
Corrected (Line 525).
Reference 22 incomplete
Corrected (Line 684 – reference 34).
